# Knowledge, Attitude and Practice of the Community General Practice Teams on Dysphagia Complicated with Aspiration Pneumonia after Stroke

**DOI:** 10.3390/healthcare11233073

**Published:** 2023-11-30

**Authors:** Daikun He, Xueting Shen, Lina Wang, Zhigang Pan

**Affiliations:** 1Department of General Practice, Zhongshan Hospital, Fudan University, Shanghai 200032, China; daikun_he@126.com (D.H.); 21111210157@m.fudan.edu.cn (X.S.); 2Department of General Practice, Jinshan Hospital, Fudan University, Shanghai 201508, China; wanglina921@alu.fudan.edu.cn; 3Center of Emergency and Critical Care Medicine, Jinshan Hospital, Fudan University, Shanghai 201508, China

**Keywords:** knowledge, attitude and practice(KAP), stroke, dysphagia, pneumonia, aspiration, physicians

## Abstract

This cross-sectional study investigated the knowledge, attitude and practice (KAP) of community general practice (GP) team members on dysphagia complicated with aspiration pneumonia after stroke in Shanghai between October 2022 and November 2022 using a self-administered questionnaire. A total of 551 questionnaires were collected (mean age: 37.59 ± 8.86 years, 443 (80.40%) females), including 383 (69.51%) physicians. The mean KAP scores were 6.30 ± 1.54 (possible range: 0–12), 40.32 ± 5.11 (possible range: 9–45), and 72.54 ± 13.99 (possible range: 18–90), respectively. Multivariable linear regression analyses suggested that attitude (Coef = 1.29, 95%CI: 1.09–1.50), and holding research funding (Coef = −2.70, 95%CI: −5.00–−0.50) were significantly associated with practice toward dysphagia complicated with aspiration pneumonia after stroke of community GP team members. The structural equation model (SEM) indicated that knowledge had a direct influence on attitude (β = 2.029, *p* < 0.001) and attitude had a direct impact on practice (β = 0.710, *p* < 0.001). Additionally, knowledge exerted both direct (β = 0.935, *p* = 0.016) and indirect effects (β = 1.442, *p* < 0.001) on practice. In conclusion, this study showed that the community GP team members had poor knowledge, favorable attitudes, and proactive practices. Education and training on the management of dysphagia complicated with aspiration pneumonia after stroke are urgently needed.

## 1. Introduction

Dysphagia is common after stroke and can result in aspiration pneumonia, poor nutrition, and dehydration [1,2]. Aspiration pneumonia refers to lung infection following oropharyngeal secretion, particulate matter, or gastric content inhalation [3,4,5]. It is estimated that 90% of patients in the intensive care unit (ICU) have at least one aspiration event during their stay [6]. The prevalence of dysphagia during the first few days after stroke is 19–65%, depending on stroke location, timing, and assessment method [1,2]. Stroke-related dysphagia resolves spontaneously in up to 50% of the affected patients but can remain a life-long issue in many patients [1,2]. The factors associated with an increased risk of dysphagia after stroke and its complications include older age, greater stroke severity and disability, poor baseline nutrition, greater lesion volume, subcortical and/or brainstem involvement, white matter involvement, dysarthria, dysphonia, cognitive impairment, and dementia [7]. In patients with stroke, dysphagia is associated with a 3.2-fold increased risk of pneumonia and aspiration is associated with an 11.6-fold increased risk of pneumonia [8]. The in-hospital mortality due to aspiration pneumonia is 11.3% in older adults (≥65 years of age) [9] and the 1-year mortality can reach 49% [10]. Hence, the authoritative societies’ guidelines on managing stroke emphasize the importance of screening and managing dysphagia during the acute stroke care period [2,11,12,13].

General practitioners (GPs) are at the forefront of community care and are often the bridge to specialized care [14]. After the acute phase, stabilization, and rehabilitation of stroke, the patients are often returned to their living community, where they are taken in charge by GP community members. GPs must be aware of the risk of dysphagia complicated with aspiration pneumonia after stroke and must be able to recognize a case of aspiration pneumonia and initiate the proper care promptly [15,16]. Some studies examined the knowledge and practice of healthcare providers, including physicians, nurses, and residents, toward dysphagia management [17,18], post-stroke dysphagia [19,20], and aspiration pneumonia [21], generally revealing low or moderate knowledge and practice. However, there are no studies on the KAP of GPs toward dysphagia complicated with aspiration pneumonia after stroke.

The knowledge, attitude, and practice (KAP) methodology is a quantitative approach that uses predefined questions formatted in standardized questionnaires and provides access to quantitative and qualitative information on a specific subject. The KAP model posits that enhanced knowledge fosters favorable attitudes, resulting in improved practices or behaviors and misconceptions or misunderstandings that may represent barriers to clinical activities [22,23]. In this study, the hypotheses were that (1) knowledge of community GP team members leads to positive attitudes toward dysphagia complicated with aspiration pneumonia after stroke, (2) good knowledge facilitates more proactive practices, and (3) good knowledge leads to more proactive practices.

Therefore, this study examined the KAP of community GP team members toward dysphagia complicated with aspiration pneumonia after stroke. The results could identify gaps that would require improvement through additional training.

## 2. Materials and Methods

### 2.1. Study Design and Participants

This cross-sectional study was conducted at Shanghai Community Health Service Center between 29 October 2022, and 22 November 2022, and enrolled physicians and nurses from GP teams. Physicians and nurses in training or internships were excluded. This study was approved by the Ethics Committee of Jinshan Hospital, Fudan University (No. JIEC 2022-S81). All participants signed the informed consent form before completing the survey.

### 2.2. Procedures

The questionnaire used in this study was based on the 2019 Updated Chinese Consensus Statement on Stroke-associated Pneumonia guidelines [24], Chinese Dysphagia and Nutrition Management Manual for Acute Stroke Patients [25], China Expert Consensus on Home Nutrition Administration for Elderly Patients with Dysphagia (version 2018), Anti-Infective Drug Therapy for Stroke Complicated by Pneumonia: Recommendations of the Consensus Study Group on Stroke Complicated by Pneumonia (complete translation), Evidence Summary of Prevention of Stroke Associated Pneumonia in Patients with Dysphagia, Construction and Implementation of the Management Program of Stroke Patients with Dysphagia-based Evidence, and a previous study [26].

Two rounds of expert consultation were conducted to ensure the quality of the questionnaire. Five experts were selected, including two GP experts, a neurologist, and two nursing experts. After consultation, three new questions were added to the knowledge dimension based on the results of expert inquiries. K7: For individuals who have suffered a stroke, the preventive use of antimicrobial medications is a viable option. K11: To reduce the risk of stroke-associated pneumonia, it is advisable to minimize the use of medications, including glucocorticoids, proton pump inhibitors, and sedatives. K12: When treating aspiration pneumonia, it is essential that antimicrobial therapy encompasses anaerobic bacteria, Staphylococcus aureus, Gram-negative rods, aerobic bacteria, and fungi.

The final questionnaire included (1) demographic data of the participants, including age, gender, level of education, type of occupation, length of service, professional title, research funding, and publications; (2) 12 questions about patients with dysphagia complicated by aspiration pneumonia after stroke (scored 1 point for correct answers and 0 points for incorrect or unclear answers, ranging from 0 to 12 points); (3) nine questions about the attitude of community GP team members towards patients with dysphagia complicated by aspiration pneumonia after stroke (5-point Likert scale, from very positive (5 points) to very negative (1 point), ranging from 9 to 45 points); and (4) 18 questions on the practice of the community GP team on patients with dysphagia complicated by aspiration pneumonia after stroke such as dysphagia screening, swallowing, rehabilitation training, aspiration prevention, and feeding pathways and other measures (5-point Likert scale, from always (5 points) to never (1 point), ranging from 18 to 90 points). In order to ensure the reliability of the questionnaire responses, a small pre-test was carried out, involving 52 copies. The pre-test results indicated high internal consistency, with a Cronbach’s α value of 0.950. A cutoff of 70% of the total score was used to determine good knowledge, favorable attitude, and proactive practice [27].

A convenience sampling approach was used to contact the directors of community health service centers and explain the importance of the survey. Once their participation was confirmed, a skilled research assistant offered a phone number or WeChat contact to stay in touch and provided relevant instructions to ensure that every valid questionnaire was accurately completed. The survey was delivered through an online platform. In order to ensure that only healthcare workers responded, the community directors personally distributed the survey to the community healthcare workers. The online questionnaire with a QR code was established using the Wen Juan Xing (WJX) platform (https://www.wjx.cn (accessed on 29 October 2022)) to collect data through WeChat. The participants logged in by scanning the QR code and completing the questionnaire. In order to ensure quality and completeness, each IP address could only be used once to submit the questionnaire. All questions had to be answered. The research team members checked all questionnaires for completeness, internal coherence, and rationality.

### 2.3. Statistical Analyses

The sample size was determined by multiplying 10 by the number of items in the questionnaire (*n* = 39) and adding 10% to account for eventual invalid responses [28]. Therefore, the minimal sample size was 429.

STATA 17.0 (Stata Corporation, College Station, TX, USA) was used for statistical analyses. Continuous data were expressed as means ± standard deviations (SD) and analyzed using ANOVA. Categorical data were expressed as *n* (%) and analyzed using the chi-squared test. Pearson’s correlation was used to analyze the correlation between the knowledge, attitude, and practice scores. Univariable and multivariable linear regression analyses were used to analyze the factors influencing practice. Variables with *p* < 0.20 in the univariable analyses of the total practice score were included in the multivariable linear regression analyses. A two-sided *p* < 0.05 was considered statistically significant.

## 3. Results

### 3.1. Characteristics of the Participants

A total of 551 questionnaires were collected (mean age: 37.59 ± 8.86 years, 443 (80.40%) females), including 383 (69.51%) physicians. The mean KAP scores were 6.30 ± 1.54 (possible range: 0–12), 40.32 ± 5.11 (possible range: 9–45), and 72.54 ± 13.99 (possible range: 18–90), respectively. There were significant differences in knowledge between different lengths of service (*p* = 0.041), different professional titles (*p* = 0.013), and with and without publications (*p* = 0.026), in attitude between different levels of education (*p* < 0.001), and in practice between GP team members with and without research funding (Table 1). There were no significant differences in knowledge, attitude, and practice between physicians and nurses other than that regarding P7 (oral hygiene) (*p* = 0.031) (Table 2).

### 3.2. Knowledge Dimension

Among the knowledge dimensions, “Dysphagia is one of the most common clinical complications after stroke, greatly increasing the risk of death and poor prognosis. (K1)” had the highest correct rate, while “Anti-infective therapy should be initiated within 4 h in all patients with suspected or confirmed stroke-associated pneumonia if patients were complicated by sepsis or septic shock. (False) (K6)” had the lowest correct rate (Appendix A).

### 3.3. Attitude and Practice Dimensions

Appendix A present the distribution of the attitude and practice dimensions, respectively. The participants responded positively to all statements in attitude (Appendix A) and practice (Appendix A).

### 3.4. Correlation Analyses

Pearson’s correlation analyses showed that the knowledge scores were correlated to the attitude scores and practice scores “Swallowing screening and rehabilitation training” item scores, “Aspirations prevent and feeding pathway” item scores, and “Other measures” item scores. The attitude scores correlated with the practice scores “Swallowing screening and rehabilitation training” item scores, “Aspirations prevent and feeding pathway” item scores, and “Other measures” item scores. The practice scores correlated with the “Swallowing screening and rehabilitation training” item scores, “Aspirations prevent and feeding pathway” item scores, and “Other measures” item scores (Table 3).

### 3.5. Univariable and Multivariable Analyses

The multivariable linear regression analyses suggested that attitude (Coef = 1.29, 95%CI: 1.09–1.50) and holding research funding (Coef = −2.70, 95%CI: −5.00–−0.50) were significantly associated with practice toward dysphagia complicated with aspiration pneumonia after stroke of community GP team members (Table 4).

Multivariable linear regression analyses showed that attitude (Coef = 0.53, 95%CI: 0.43–0.62), nurses (Coef = 1.58, 95%CI: 0.39–2.77), length of service above 20 years (Coef = −3.50, 95%CI: −6.35–−0.65), holding research funding (Coef = −1.84, 95%CI: −3.06–−0.61), and having publications (Coef = 1.48, 95%CI: 0.30–2.66) were significantly associated with the practice of “Swallowing screening and rehabilitation training”. Attitude (Coef = 0.34, 95%CI: 0.28–0.41) and holding research funding (Coef = −1.44, 95%CI: −2.33–−0.56) were significantly associated with the practice of “Aspiration prevention and feeding pathway”. Additionally, attitude (Coef = 0.40, 95%CI: 0.33–0.47), professional title of associate senior/senior (Coef = 2.69, %CI: 0.31–5.06), and holding research funding (Coef = −1.37, 95%CI: −2.29–−0.44) were significantly associated with the practice of “Other measures (including eating position, food form, compensatory strategies, eating speed, and oral hygiene)” (Table 5).

### 3.6. Structural Equation Model

The structural equation model (SEM) indicated that knowledge had a direct influence on attitude (β = 2.029, *p* < 0.001) and attitude had a direct impact on practice (β = 0.710, *p* < 0.001). Additionally, knowledge exerted both direct (β = 0.935, *p* = 0.016) and indirect effects (β = 1.442, *p* < 0.001) on practice (Appendix A).

## 4. Discussion

This study found that knowledge of community GP team members was positively correlated with their attitude and practice toward dysphagia complicated with aspiration pneumonia after stroke, and knowledge and attitude were positively correlated with their practice, which verified our hypotheses. However, holding research funds was negatively correlated with their practice. Therefore, healthcare professionals need education and training on dysphagia management to promote proactive practices, and senior staff members should be involved in dysphagia management to improve the quality of care.

### 4.1. Knowledge

The present study suggested that the community GP team members had moderate knowledge of dysphagia complicated by aspiration pneumonia after stroke. Similarly, Sanchez-Sanchez et al. [17] reported that the KAP of dysphagia was moderate-to-low among healthcare providers in Spain (including 24.6% working in internal medicine, 10.9% in geriatrics, 9.9% in primary healthcare, and 7.9% in oncology). In Sichuan (China), Luo et al. [18] reported that although the dysphagia attitudes and practices of geriatrics nurses were acceptable, their knowledge was low. More specifically, in post-stroke dysphagia, a study in Wuhan (China) showed that the KAP of nurses (working in neurology, neurosurgery, rehabilitation, or geriatrics departments) was low, especially in lower-grade hospitals [19]; similar results were observed in Beijing of neurological nurses [29]. Similar results regarding aspiration pneumonia KAP were observed for neurological nurses in Burkina Faso [20] and among nurses and residents in Saudi Arabia [21]. The present study identified specific knowledge items that would require training in Shanghai, including the definition of aspiration pneumonia, the timing of antibiotics, the complications of dysphagia and aspiration, and the etiology of aspiration.

### 4.2. Relationship between Knowledge, Attitude, and Practice

The KAP theory considers that knowledge acquisition, attitude generation, and behavior formation are all interrelated and affect each other [22,23]. According to the KAP theory, knowledge is the basis of practice [22,23]. Knowledge is the intellectual prerequisite for forming a correct behavior, and knowledge also indirectly affects behavior by affecting attitude [30,31,32]. Attitude will also influence how a correct behavior is performed [30,31,32]. In the present study, positive correlations were observed among the knowledge, attitude, practice, and specific practice scores. The working hypotheses of the SEM were elaborated along with the KAP theory, and the SEM model also supported these relationships. Indeed, in the SEM, knowledge directly influenced attitude and practice, and attitude directly influenced practice. Hence, knowledge had direct and indirect (through attitude) influences on practice.

Three previous studies in China showed that the hospital level was the main factor affecting the KAP of post-stroke dysphagia in nurses [18,19,29]. In the multivariable analyses, the knowledge and attitude scores were associated with better practice scores. When considering the three specific practice items selected for analyses, only the knowledge and attitude scores were consistently independently associated with the scores of the three practice items. Hence, in the study population, only knowledge and attitude can be changed to improve the practice of aspiration pneumonia prevention in stroke patients. Higher professional titles are often associated with less direct work with the patients in the wards, leaving more menial work to the younger professionals. Still, less contact with the patients can decrease KAP in specific areas. Indeed, a study showed a negative association between the nurse–patient ratio and the quality of nursing care [33].

The study reveals a negative association between healthcare professionals holding research funding and their practice of dysphagia complicated with aspiration pneumonia after stroke. Several potential factors may contribute to this association. Firstly, excessive work pressure may divert the attention of professionals with research funding from the specific clinical practice. Secondly, inadequate resources, including manpower and financial support, may hinder effective engagement in dysphagia prevention and management. Thirdly, issues related to management and coordination further impact healthcare professionals’ ability to actively participate in clinical practice, affecting their focus on daily medical duties. In summary, addressing excessive work pressure, resource shortages, and improving project management could enhance the practice levels among healthcare professionals holding research funding in managing dysphagia complications post-stroke. The observed significant association between having publications and the practice of “swallowing screening and rehabilitation training” may be attributed to increased expertise and knowledge transfer, expanded professional networks fostering collaboration, and a commitment to continuous learning among individuals with a publication record.

To address these issues, the following concrete measures are proposed: firstly, ensure effective clinical practice by strategically distributing human, material, and financial resources to support healthcare professionals with research funding. Simultaneously, provide targeted professional training to enhance their skills and proficiency. Secondly, foster a focused clinical approach by streamlining workflows and increasing human resources. These strategies reduce the burden on professionals with research funding, allowing for enhanced attention to clinical responsibilities. Thirdly, facilitating active participation in projects without compromising daily medical duties, establishing a collaborative team atmosphere, providing necessary support, and fostering effective communication will help ensure the seamless integration of project tasks with routine clinical work.

### 4.3. Future Directions

Using the results of the present study, educational interventions should be designed to improve the knowledge of community GP team members. Indeed, improving knowledge will ameliorate the KAP toward post-stroke dysphagia [29]. Stakeholders and hospital managers should be aware of the gaps in KAP in their institution and take the necessary measures to improve the situation. Specifically, training on recognizing dysphagia, the appropriate swallowing postures and maneuvers, and diet modifications should be strengthened [34,35].

### 4.4. Limitations

There are some limitations in this study. Firstly, this is a single-center study, which only represents a specific area (Shanghai) and it may not be generalized to other areas or subjects [22,23]. Secondly, although a KAP study allows for identifying specific subjects or items to be improved, the survey did not provide how to improve those points, and the results only apply to the covered points. Finally, whether the participants were engaging in managing patients with stroke was not inquired.

## 5. Conclusions

The results showed that the knowledge about dysphagia complicated with aspiration pneumonia after stroke of community GP team members was positively correlated with their attitude and practice toward dysphagia complicated with aspiration pneumonia after stroke. In addition, knowledge and attitude were positively correlated with their practice, while holding research funding was negatively associated with their practice. The findings of this study may help the development of educational interventions to improve the management of dysphagia after stroke and reduce the risk of aspiration pneumonia.

## Figures and Tables

**Table 1 healthcare-11-03073-t001:** Baseline characteristics and KAP scores.

Variables	*n* = 551	Knowledge Scores		Attitude Scores		Practice Scores	
		Mean ± SD	*p*	Mean ± SD	*p*	Mean ± SD	*p*
**Total scores**		6.30 ± 1.54	-	40.32 ± 5.11	-	72.54 ± 13.99	-
**Age (Mean ± SD)**	37.59 ± 8.86	-	-	-	-	-	-
**Gender**			0.318		0.297		0.597
Male	108 (19.60)	6.17 ± 1.43		39.86 ± 5.08		71.90 ± 13.02	
Female	443 (80.40)	6.33 ± 1.56		40.43 ± 5.12		72.69 ± 14.22	
**Level of education**			0.326		<0.001		0.481
Vocational education and below	103 (18.69)	6.17 ± 1.62		38.82 ± 5.55		71.66 ± 13.61	
Bachelor and above	448 (81.31)	6.33 ± 1.52		40.67 ± 4.95		72.74 ± 14.08	
**Type of occupation**			0.661		0.690		0.073
Physician	383 (69.51)	6.32 ± 1.48		40.26 ± 5.01		71.83 ± 14.45	
Nurse	168 (30.49)	6.26 ± 1.67		40.45 ± 5.35		74.15 ± 12.77	
**Length of service**			0.041		0.381		0.543
<5 years	93 (16.88)	6.05 ± 1.58		39.86 ± 4.83		72.10 ± 15.04	
5–10 years	144 (26.13)	6.17 ± 1.53		40.60 ± 5.38		73.93 ± 12.57	
10–20 years	141 (25.59)	6.28 ± 1.52		40.77 ± 4.98		72.43 ± 13.66	
>20 years	173 (31.40)	6.55 ± 1.52		39.97 ± 5.13		71.70 ± 14.78	
**Professional title**			0.013		0.759		0.258
None	34 (6.17)	6.24 ± 1.72		39.71 ± 5.21		71.76 ± 13.68	
Junior	178 (32.30)	6.04 ± 1.61		40.61 ± 5.10		73.61 ± 13.58	
Intermediate grade	283 (51.36)	6.38 ± 1.46		40.21 ± 5.13		71.52 ± 14.56	
Associate senior/senior	56 (10.16)	6.75 ± 1.48		40.34 ± 5.05		74.73 ± 12.21	
**Research funding**			0.826		0.810		0.025
Yes	159 (28.86)	6.28 ± 1.62		40.24 ± 5.42		70.44 ± 14.78	
No	392 (71.14)	6.31 ± 1.51		40.35 ± 4.99		73.39 ± 13.58	
**Publications**			0.026		0.250		0.230
Yes	241 (43.74)	6.46 ± 1.50		40.61 ± 5.15		73.35 ± 13.34	
No	310 (56.26)	6.17 ± 1.56		40.10 ± 5.08		71.91 ± 14.46	

**Table 2 healthcare-11-03073-t002:** Comparison of the KAP between physicians and nurses.

Factor or Statement (Mean ± SD)	Participants (*n* = 551)		*p*
	Physician	Nurse	
**Knowledge**	6.32 ± 1.48	6.26 ± 1.67	0.661
**Attitude**			
Total scores	40.26 ± 5.01	40.45 ± 5.35	0.690
Learning and training (A1/A8)	8.93 ± 1.17	8.87 ± 1.24	0.553
Prevention, screening, and education (A2/A5/A7/A9)	18.14 ± 2.10	18.18 ± 2.36	0.863
Community management (A3/A4/A6)	13.19 ± 2.02	13.40 ± 2.02	0.241
**Practice**			
Total scores	71.83 ± 14.45	74.15 ± 12.77	0.073
Swallowing screening and rehabilitation training (P1.1–P1.7)	27.20 ± 6.51	28.30 ± 5.54	0.056
Aspirations prevention and feeding pathways (P2.1–P2.5)	19.50 ± 4.42	19.99 ± 4.25	0.226
Eating position (P3.1–P3.2)	8.37 ± 1.73	8.65 ± 1.48	0.065
Food form (P4)	4.20 ± 0.84	4.26 ± 0.86	0.417
Compensatory strategies (P5)	4.13 ± 0.95	4.25 ± 0.86	0.145
Eating speed (P6)	4.26 ± 0.86	4.33 ± 0.89	0.351
Oral hygiene (P7)	4.18 ± 0.93	4.36 ± 0.83	0.031

**Table 3 healthcare-11-03073-t003:** Pearson’s correlation analyses.

	Knowledge	Attitude	Practice	Swallowing Screening and Rehabilitation Training	Aspirations Prevent and Feeding Pathway	Other Measures
**Knowledge**	1					
**Attitude**	0.26 *	1				
**Practice**	0.20 *	0.50 *	1			
Swallowing screening and rehabilitation training	0.17 *	0.46 *	0.93 *	1		
Aspirations prevent and feeding pathway	0.18 *	0.43 *	0.92 *	0.80 *	1	
Other measures	0.21 *	0.46 *	0.88 *	0.70 *	0.76 *	1

* *p* < 0.001.

**Table 4 healthcare-11-03073-t004:** Univariable and multivariable linear regression analyses on practice.

	Univariable Linear Regression	Multivariable Linear Regression
	Coef (95%CI)	Coef (95%CI)
		R^2^ = 0.2550
		F = 63.75 (*p* = 0.021)
**Knowledge score**	1.84 (1.10, 2.59)	0.71 (0.03, 1.39)
**Attitude score**	1.35 (1.15, 1.55)	1.29 (1.09, 1.50)
**Age**	−0.0 (−0.1, 0.08)	
**Gender**	Ref Male	
Female	0.79 (−2.1, 3.74)	
**Level of education**	Ref Vocational education and below	
Bachelor and above	1.07 (−1.9, 4.08)	
**Type of occupation**	Ref Physician	
Nurse	2.31 (−0.2, 4.85)	
**Length of service**	Ref < 5 years	
5–10 years	1.83 (−1.8, 5.49)	
10–20 years	0.33 (−3.3, 4.00)	
>20 years	−0.3 (−3.9, 3.13)	
**Professional title**	Ref None	
Junior	1.84 (−3.2, 6.98)	
Intermediate grade	−0.2 (−5.2, 4.73)	
Associate senior/senior	2.96 (−2.9, 8.93)	
**Research funding**	Ref No	Ref No
Yes	−2.9 (−5.5, −0.3)	−2.70 (−5.00, −0.50)
**Publications**	Ref No	
Yes	1.44 (−0.9, 3.80)	

**Table 5 healthcare-11-03073-t005:** Multivariable linear regression analyses.

Factors	Swallowing Screening and Rehabilitation Training	Aspirations Prevention and Feeding Pathways	Other Measures
	Coef (95%CI)	Coef (95%CI)	Coef (95%CI)
	*R*^2^ = 0.2393	*R*^2^ = 0.1938	*R*^2^ = 0.2282
	F= 13.36 (*p* < 0.001)	F= 10.44 (*p* < 0.001)	F = 12.61 (*p* < 0.001)
**Knowledge score**	0.25 (−0.00, 0.56)	0.20 (−0.00, 0.43)	0.23 (−0.00, 0.47)
**Attitude score**	0.53 (0.43, 0.62)	0.34 (0.28, 0.41)	0.40 (0.33, 0.47)
**Age**	0.11 (−0.00, 0.23)	0.00 (−0.08, 0.09)	−0.02 (−0.11, 0.07)
**Gender**	Ref Male		
Female	−0.53 (−1.76, 0.69)	−0.26 (−1.14, 0.62)	−0.21 (−1.13, 0.71)
**Level of education**	Ref Vocational education and below		
Bachelor and above	−0.26 (−1.66, 1.13)	−0.58 (−1.59, 0.42)	−0.49 (−1.54, 0.56)
**Type of occupation**	Ref Physician		
Nurse	1.58 (0.39, 2.77)	0.48 (−0.38, 1.33)	0.78 (−0.11, 1.68)
**Length of service**	Ref <5 years		
5–10 years	−0.09 (−1.73, 1.56)	0.12 (−1.06, 1.31)	−0.17 (−1.42, 1.07)
10–20 years	−1.83 (−3.87, 0.20)	−0.68 (−2.15, 0.78)	−0.62 (−2.16, 0.91)
>20 years	−3.50 (−6.35, −0.65)	−0.76 (−2.81, 1.29)	−0.24 (−2.40, 1.90)
**Professional title**	Ref None		
Junior	0.61 (−1.65, 2.87)	0.27 (−1.36, 1.90)	0.88 (−0.82, 2.60)
Intermediate grade	0.16 (−2.31, 2.64)	0.19 (−1.60, 1.97)	1.21 (−0.66, 3.09)
Associate senior/senior	2.07 (−1.07, 5.21)	1.33 (−0.93, 3.59)	2.69 (0.31, 5.06)
**Research funding**	Ref No		
Yes	−1.84 (−3.06, −0.61)	−1.44 (−2.33, −0.56)	−1.37 (−2.29, −0.44)
**Publications**	Ref No		
Yes	1.48 (0.30, 2.66)	0.62 (−0.23, 1.47)	0.64 (−0.25, 1.53)

## Data Availability

The data presented in this study are available in the article.

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
