# Peer review of "Knowledge, Attitude and Practice of the Community General Practice Teams on Dysphagia Complicated with Aspiration Pneumonia after Stroke"

_healthcare, 2023, doi:10.3390/healthcare11233073_

Round 1

Reviewer 1 Report

Comments and Suggestions for Authors

Thank you for your effort in preparing this manuscript. I think this paper has great potential but is not ready in its current form.

The overall hypothesis of the paper is not clear.

The presentation of results is highly detailed which confuses the interpretation as a reader. It is not necessary to repeat information in the text and in the tables. I think the results could be clearer and condensed according to your hypothesis and research questions.

The discussion paper would be enhanced by the use of subheadings and themes according to the findings in your results. The discussion needs clearer linkages to your results, as well as more emphasis about the clinical significance of the study.

Comments on the Quality of English Language

The manuscript requires grammatical revisions which I am happy to comment on in more detail. At this stage, the overall presentation of the results and discussion require important improvements.

Author Response

The overall hypothesis of the paper is not clear.

       Response: We thank the Reviewer. We clarified the hypothesis in the Introduction. In this study, the hypotheses were that 1) adequate knowledge of community GP team members leads to positive attitudes toward dysphagia and aspiration pneumonia after stroke, 2) good knowledge facilitates more proactive practices, and 3) good knowledge leads to more proactive practices.

The presentation of results is highly detailed which confuses the interpretation as a reader. It is not necessary to repeat information in the text and in the tables. I think the results could be clearer and condensed according to your hypothesis and research questions.

       Response: We thank the Reviewer. We simplified the description of the results. We kept the details in the figures and tables and only wrote about the important results.

The discussion paper would be enhanced by the use of subheadings and themes according to the findings in your results. The discussion needs clearer linkages to your results, as well as more emphasis about the clinical significance of the study.

       Response: We thank the Reviewer for the comments. We revised the Discussion and added subheadings as suggested. The results have limited clinical implications since the study examined the KAP of healthcare providers. Nevertheless, using the results of the present study, educational interventions should be designed to improve the knowledge of community GP team members. Indeed, improving knowledge should improve the KAP toward post-stroke dysphagia [1]. Stakeholders and hospital managers should be aware of the gaps in KAP in their institution and take the necessary measures to improve the situation. Specifically, training on recognizing dysphagia, the appropriate swallowing postures and maneuvers, and diet modifications should be strengthened [2, 3].

Comments on the Quality of English Language

The manuscript requires grammatical revisions which I am happy to comment on in more detail. At this stage, the overall presentation of the results and discussion require important improvements.

       Response: The manuscript was proofread to improve the presentation of the results and discussion.

Reviewer 2 Report

Comments and Suggestions for Authors

Dear Authors,

I read your work entitled “Knowledge, attitude, and practice of the community general practice teams on dysphagia complicated by aspiration pneumonia after stroke” and here I enclose my recommendations to you:

1.     There is a need for editing of English language errors. Please, have a more thorough “look” in the text for syntax and terminology issues.

2.     The “Introduction” section lacks information and does not have a clear rational. The text does not lead “smoothly” to the aims of the study. I suggest the Authors to address those issues.

3.     The “Methods” are sound and readers friendly. The Authors state that A7, A11, A12 questions were reviewed and changed. I suggest the Authors, if that is possible, to have a table of the old A7, A11, A12 questions in contrast to the reviewed ones.

4.     The “Results” are readers friendly and I congratulated the Authors for that.

5.     The “Discussion” section pg. 9 lines 185-190 how this is connected to the current study? If it does, I suggest Authors to connect it properly. In pg. 10, lines 212-230 the authors have no adequate reference to support their text. Please, address that as well.

Thank you.

Comments on the Quality of English Language

1) Moderate editing of English language required

2) Syntax issues throughout the manuscript

3) Terminology issues 

Author Response

There is a need for editing of English language errors. Please, have a more thorough “look” in the text for syntax and terminology issues.

       Response: We thank the Reviewer for the comment. We proofread the manuscript to improve the grammar and flow.

The “Introduction” section lacks information and does not have a clear rational. The text does not lead “smoothly” to the aims of the study. I suggest the Authors to address those issues.

       Response: We thank the Reviewer. We revised the Introduction to improve the flow and rationale.

The “Methods” are sound and readers friendly. The Authors state that A7, A11, A12 questions were reviewed and changed. I suggest the Authors, if that is possible, to have a table of the old A7, A11, A12 questions in contrast to the reviewed ones.

       Response: We thank the Reviewer for the comments. Regarding A7, A11, and A12, they were added after the experts suggested them. They were absent from the original version. It was clarified in the manuscript.

The “Results” are readers friendly and I congratulated the Authors for that.

       Response: We thank the Reviewer for the comment.

The “Discussion” section pg. 9 lines 185-190 how this is connected to the current study? If it does, I suggest Authors to connect it properly. In pg. 10, lines 212-230 the authors have no adequate reference to support their text. Please, address that as well.

       Response: We thank the Reviewer. We have modified the description of The "Discussion" section pg. 9 lines 185-190 based on the results of the supplementary SEM. We have made modifications to the discussion section In pg. 10, lines 212-230 based on your suggestion.

Reviewer 3 Report

Comments and Suggestions for Authors

The authors conducted a questionnaire survey of local doctors and nurses regarding aspiration pneumonia after stroke. Based on these results, they were able to grasp the level of knowledge and practice that the subjects had. And because of the correlation between knowledge scores, attitude scores, and practice scores, the authors argue for the need for training. However, I have doubts about the authors' assertion that one cannot put it into practice due to lack of knowledge or attitude (or that one can put it into practice by acquiring these).

If doctors and nurses who have had the opportunity to examine patients with stroke-related pneumonia on a regular basis (their practice scores are high), their knowledge and attitude should naturally be high. Conversely, doctors and nurses who do not have the opportunity to examine patients with stroke-related pneumonia on a regular basis (their practice scores are low) naturally have poor knowledge about stroke-related pneumonia. These environmental factors cannot be ignored. In particular, due to the characteristics of stroke-related pneumonia, whether or not the subject works in an acute care hospital may have a significant impact. However, in this survey, it is unclear what type of medical care hospitals the subjects work at. Therefore, we do not know whether low practice scores are due to a lack of knowledge and training, or simply a lack of opportunity to deal with such patients.

Finally, based on these results, the authors claim that “Community general practice teams had moderate knowledge, positive attitude, and proactive practice.” However, the passing standard score that is the basis for the authors' judgment is not specified. What criteria did you use to arrive at this decision?

Another small issue is that the table numbers are incorrect so please check them.

Author Response

The authors conducted a questionnaire survey of local doctors and nurses regarding aspiration pneumonia after stroke. Based on these results, they were able to grasp the level of knowledge and practice that the subjects had. And because of the correlation between knowledge scores, attitude scores, and practice scores, the authors argue for the need for training. However, I have doubts about the authors' assertion that one cannot put it into practice due to lack of knowledge or attitude (or that one can put it into practice by acquiring these).

       Response: We thank the Reviewer for the comment. According to the KAP theory, knowledge is the basis for practice, while attitude is the force driving practice [4, 5]. Adequate practice requires the knowledge of how to perform a given action adequately. Experience will lead to knowledge that will support practice. We agree that the KAP theory is only a construct and that it is imperfect in its representation of reality. Nevertheless, it is true that adequate practice requires adequate knowledge of the actions to be performed. One could argue that experience also improves practice, but experience brings a form of knowledge that is different from academic knowledge but is still knowledge. The present study could not differentiate between the two forms of knowledge. Of course, knowledge is a dynamic construct that evolves with time due to educational activities (e.g., conferences and continuing education) and experience.

If doctors and nurses who have had the opportunity to examine patients with stroke-related pneumonia on a regular basis (their practice scores are high), their knowledge and attitude should naturally be high. Conversely, doctors and nurses who do not have the opportunity to examine patients with stroke-related pneumonia on a regular basis (their practice scores are low) naturally have poor knowledge about stroke-related pneumonia. These environmental factors cannot be ignored. In particular, due to the characteristics of stroke-related pneumonia, whether or not the subject works in an acute care hospital may have a significant impact. However, in this survey, it is unclear what type of medical care hospitals the subjects work at. Therefore, we do not know whether low practice scores are due to a lack of knowledge and training, or simply a lack of opportunity to deal with such patients.

       Response: We agree with the Reviewer. The present study included GPs from the Shanghai area. Whether or not they were actively engaged in the care of patients with stroke was not examined. We considered that any GP working in the community was susceptible to encountering patients with stroke at one point or another in his/her career. Hence, all GPs could participate. Moreover, we supplemented the findings of the structural equation model (SEM) in the results section, which can demonstrate the direction between knowledge, attitude, and practice.

Finally, based on these results, the authors claim that “Community general practice teams had moderate knowledge, positive attitude, and proactive practice.” However, the passing standard score that is the basis for the authors' judgment is not specified. What criteria did you use to arrive at this decision?

       Response: We thank the Reviewer. We use a self-administered questionnaire to assess knowledge, attitude, and practice, with scores ≥70% indicating good knowledge, favorable attitude, and proactive practice. The mean knowledge, attitude, and practice scores were 6.30±1.54 (11 items, possible scores of 0-11), 40.32±5.11 (9 items, possible scores of 9-45), and 72.54±13.99 (18 items, possible scores of 19-90), respectively. The cut-off values of knowledge, attitude and practice are respectively 7.7, 31.5, and 63.0 point.

Another small issue is that the table numbers are incorrect so please check them.

       Response: We thank the Reviewer. We verified the numbering of the tables.

Reviewer 4 Report

Comments and Suggestions for Authors

Thank you for your study on knowledge, attitudes and practice of post-stroke dysphagia and aspiration pneumonia among community healthcare workers. The study is important, however the manuscript will require judicious reformatting before it is acceptable for publication. Please see detailed comments/recommendations below. 

ABSTRACT:

Detailed information is presented without context. For example, what is the meaning of the listed score range- low knowledge to high knowledge? 

In general, I recommend rewriting the Abstract so that it provides a big picture overview of the study. When listing detailed results please first provide adequate context. 

What does "having projects" mean? 

What do you mean by "Aspirations prevent and feeding pathway?" This phrase doesn't make sense. 

INTRODUCTION:

The first sentence needs work. The detailed information in parentheses is awkward and reduces readability. I recommend expanding this into 2-3 sentences, starting with something simple (for example: "Impaired swallowing is common after stroke and can result in aspiration pneumonia, poor nutrition, and dehydration") and adding the detailed information in subsequent sentences. 

When referencing the presence of dysphagia and aspiration after stroke, can you use another index of frequency besides relative risk- this is overly technical and not intuitive to the reader. Percentage, incidence, prevalence, or fold difference would be better. 

When listing factors associated with increased risk of dysphagia, remove reduced max pitch- this is redundant as this is implied with dysphonia which is listed previously. 

How does your KAP survey compare to those previously published on stroke, dysphagia, and aspiration? Is your survey adapted from these earlier publications or completely novel? 

RESULTS:

The results are disorganized and difficult to comprehend, but this can be resolved with careful reformatting and organization. 

First, in the text portion of the results:

-Eliminate redundancy; you do not need to list values in the text that are also shown in a table

-Break up the Results text into subtitled sections. The subheading should indicate to the reader which aspect of the results are being reported in that particular section. 

Next, the tables need a lot of work:

-In the current manuscript, I count 8 tables, but the last table is titled Table 6. There are two Table 1s and Two Table 2s. Please check that tables are appropriately named and remove redundancy. 

-The exhaustive nature of the tables hinders readability. It would be more helpful to see fewer tables featuring relevant results than a run-on table with every comparison generated by your statistical tests. I recommend condensing your tables by focusing on statistically significant findings and/or clinically relevant points. 

-I recommend titling tables with a declarative statement that summarizes what is shown in the table

The figures also need work:

-Each figure should have a text legend below the figure that provides context for what is shown 

-The figures are too small to read. Enlarge each figure so that the text is of readable size in the manuscript. 

-Consider presenting the items of your survey in a uniform manner. You may find that adding a figure showing the knowledge portion of the survey and plots % of true/false responses (similar to your plots for the attitude and practice portions) improves cohesion. 

-For the knowledge portion of the survey, I recommend indicating to the reader whether each statement is true or false. Perhaps by a parenthetical (t) or (f) after each statement is listed.

DISCUSSION:

Expanded discussion of results would be appropriate. Currently, discussion of the data reads like a list of statistical outcomes which was already done in the Results section. What are the main findings and take home message? 

Comments on the Quality of English Language

The English is comprehensible overall.  Line 23 of the Abstract is one of few instances where meaning is obscured by incorrect grammar: "Aspirations prevent and feeding pathway." Did you mean "Aspiration prevention and feeding pathway?" Please proofread and edit language as needed. 

Author Response

ABSTRACT:

Detailed information is presented without context. For example, what is the meaning of the listed score range- low knowledge to high knowledge? 

In general, I recommend rewriting the Abstract so that it provides a big picture overview of the study. When listing detailed results please first provide adequate context. 

       Response: We thank the Reviewer. We revised the Abstract accordingly.

What does "having projects" mean? 

       Response: We meant holding national research funding. It was revised in the manuscript.

What do you mean by "Aspirations prevent and feeding pathway?" This phrase doesn't make sense. 

       Response: We thank the Reviewer. It is the name of the practice item that was analyzed. Still, we agree that the translation was not optimal and changed it to “Aspiration prevention and feeding pathways”.

INTRODUCTION:

The first sentence needs work. The detailed information in parentheses is awkward and reduces readability. I recommend expanding this into 2-3 sentences, starting with something simple (for example: "Impaired swallowing is common after stroke and can result in aspiration pneumonia, poor nutrition, and dehydration") and adding the detailed information in subsequent sentences. 

When referencing the presence of dysphagia and aspiration after stroke, can you use another index of frequency besides relative risk- this is overly technical and not intuitive to the reader. Percentage, incidence, prevalence, or fold difference would be better. 

When listing factors associated with increased risk of dysphagia, remove reduced max pitch- this is redundant as this is implied with dysphonia which is listed previously. 

How does your KAP survey compare to those previously published on stroke, dysphagia, and aspiration? Is your survey adapted from these earlier publications or completely novel? 

       Response: We thank the Reviewer for the comments. We revised the Introduction accordingly. We added the suggested statement as an entry, followed by detailed information. The index of frequency was changed. The risk factors were revised as suggested. Some studies examined the knowledge and practice of healthcare providers, including physicians, nurses, and residents, toward dysphagia management [6, 7], post-stroke dysphagia [8, 9], and aspiration pneumonia [10], generally revealing low or moderate knowledge and practice. No study examined the KAP of GPs toward post-stroke dysphagia and aspiration pneumonia.

RESULTS:

The results are disorganized and difficult to comprehend, but this can be resolved with careful reformatting and organization. 

First, in the text portion of the results:

-Eliminate redundancy; you do not need to list values in the text that are also shown in a table

    Response: We thank the Reviewer for the comments. We have eliminate the results.

-Break up the Results text into subtitled sections. The subheading should indicate to the reader which aspect of the results are being reported in that particular section. 

Response: We thank the Reviewer for the comments. We have added subheadings in the results.

Next, the tables need a lot of work:

-In the current manuscript, I count 8 tables, but the last table is titled Table 6. There are two Table 1s and Two Table 2s. Please check that tables are appropriately named and remove redundancy. 

Response: We thank the Reviewer for the comments. We have check and modify the tables number.

-The exhaustive nature of the tables hinders readability. It would be more helpful to see fewer tables featuring relevant results than a run-on table with every comparison generated by your statistical tests. I recommend condensing your tables by focusing on statistically significant findings and/or clinically relevant points. 

Response: We thank the Reviewer for the comments. We have streamlined the description of the results section and hope it can help the readers. However, we have not made numerous modifications to the tables at present. Please understand. Thank you.

-I recommend titling tables with a declarative statement that summarizes what is shown in the table

Response: We thank the Reviewer for the comments. We have refine the tables title.

The figures also need work:

-Each figure should have a text legend below the figure that provides context for what is shown 

-The figures are too small to read. Enlarge each figure so that the text is of readable size in the manuscript. 

Response: We thank the Reviewer for the comments. We have redone only Figures 1 and 2 to improve clarity.

-Consider presenting the items of your survey in a uniform manner. You may find that adding a figure showing the knowledge portion of the survey and plots % of true/false responses (similar to your plots for the attitude and practice portions) improves cohesion. 

Response: We thank the Reviewer for the comments. Due to the limited number of options in the knowledge dimension, with only right and wrong options, we still present them in a table format. Looking forward to your further suggestions, thank you.

-For the knowledge portion of the survey, I recommend indicating to the reader whether each statement is true or false. Perhaps by a parenthetical (t) or (f) after each statement is listed.

       Response: We thank the Reviewer for the comments, we have added true or false annotations in Table 3.

DISCUSSION:

Expanded discussion of results would be appropriate. Currently, discussion of the data reads like a list of statistical outcomes which was already done in the Results section. What are the main findings and take home message? 

       Response: We thank the Reviewer. We reorganized and revised the Discussion. This study suggests that knowledge of community GP team members is positively correlated with their attitude and practice toward dysphagia and aspiration pneumonia in patients with stroke, attitude is positively correlated with their practice. Healthcare professionals need education and training on dysphagia management to promote proactive practices, and senior staff members should be involved in dysphagia management to improve the quality of care.

Round 2

Reviewer 1 Report

Comments and Suggestions for Authors

Thank you for your revisions and the opportunity to reread your manuscript, which has been positively transformed. I really appreciate the effort taken to rework this piece.

It is great to read clear hypotheses and a concise introduction. Subheadings have significantly improved the structure and flow of the results and discussion sections. Thank you. I think this study has great international clinical relevance for health professionals supporting people who have had a stroke, particularly those who are seeking community support from GP and of course have dysphagia

My main query for revision is the discussion of your hypotheses. Page 9, line 224. The results section states ‘as hypothesised’. I think this interpretation should be moved into the discussion section and made more explicit as an analysis of hypotheses. In my opinion, the discussion is the place to analyse hypotheses, rather than the results. This analysis is not clear in the first paragraph of your discussion. 

I find this paper a bit overwhelmed by tables and figures. Can any of these be removed or simplified? Are they all necessary to reflect the aims of this study? 

Page 7, is Figure 3 necessary? Perhaps some tables or figures could be provided as supplementary files, rather than as part of the main article?

Comments on the Quality of English Language

Overall high quality.

Page 1, line 41 - the opening of your introduction is noteworthy to the reader. I would consider rewording ‘large amount of colonised oropharyngeal material’. For example, is ‘large amount’ necessary? 'Colonised oropharyngeal material' also seems like it could be worded in a more natural way - it may be an English translation issue.

Page 10, line 240 typo ‘Knowledge’

Author Response

Thank you for your revisions and the opportunity to reread your manuscript, which has been positively transformed. I really appreciate the effort taken to rework this piece.

It is great to read clear hypotheses and a concise introduction. Subheadings have significantly improved the structure and flow of the results and discussion sections. Thank you. I think this study has great international clinical relevance for health professionals supporting people who have had a stroke, particularly those who are seeking community support from GP and of course have dysphagia

My main query for revision is the discussion of your hypotheses. Page 9, line 224. The results section states ‘as hypothesised’. I think this interpretation should be moved into the discussion section and made more explicit as an analysis of hypotheses. In my opinion, the discussion is the place to analyse hypotheses, rather than the results. This analysis is not clear in the first paragraph of your discussion. 

Response: We thank the Reviewer for the comments. The description of the results has been modified, and the first paragraph of the discussion is supplemented.

I find this paper a bit overwhelmed by tables and figures. Can any of these be removed or simplified? Are they all necessary to reflect the aims of this study? 

Response: We thank the Reviewer. The original Table 3 and Figures 1-3 were moved to the Supplementary Materials.

Page 7, is Figure 3 necessary? Perhaps some tables or figures could be provided as supplementary files, rather than as part of the main article?

Response: We thank the Reviewer. The original Table 3 and Figures 1-3 were moved to the Supplementary Materials.

Comments on the Quality of English Language

Overall high quality.

Page 1, line 41 - the opening of your introduction is noteworthy to the reader. I would consider rewording ‘large amount of colonised oropharyngeal material’. For example, is ‘large amount’ necessary? 'Colonised oropharyngeal material' also seems like it could be worded in a more natural way - it may be an English translation issue.

Response: We agree with the Reviewer. It was changed to “oropharyngeal secretions, particulate matter, or gastric content”.

Page 10, line 240 typo ‘Knowledge’

Response: It was corrected.

Reviewer 3 Report

Comments and Suggestions for Authors

I understand that the authors' claims are based on the KAP theory, and that there are limitations to this research. Finally, I would like to suggest that the authors present the research (future prospects) that will be needed to prove their hypothesis.

Author Response

I understand that the authors' claims are based on the KAP theory, and that there are limitations to this research. Finally, I would like to suggest that the authors present the research (future prospects) that will be needed to prove their hypothesis.

       Response: We thank the Reviewer for the comment. This study was based on the following hypotheses: 1) adequate knowledge of community GP team members leads to positive attitudes toward dysphagia and aspiration pneumonia after stroke, 2) good knowledge facilitates more proactive practices, and 3) good knowledge leads to more proactive practices. These hypotheses were verified in the present study. Indeed, the knowledge of community GP team members was positively correlated with their attitude and practice toward dysphagia and aspiration pneumonia in patients with stroke, and attitude was positively correlated with their practice. Nevertheless, healthcare professionals need education and training on dysphagia management to promote proactive practices, and senior staff members should be involved in dysphagia management to improve the quality of care. The future prospects are to develop educational activities based on the issues identified in the present study, implement them, and observe how KAP was changed in the professionals. Using the results of the present study, educational interventions should be designed to improve the knowledge of community GP team members. Indeed, improving knowledge should improve the KAP toward post-stroke dysphagia [1]. Stakeholders and hospital managers should be aware of the gaps in KAP in their institution and take the necessary measures to improve the situation. Specifically, training on recognizing dysphagia, the appropriate swallowing postures and maneuvers, and diet modifications should be strengthened [2, 3].

Reviewer 4 Report

Comments and Suggestions for Authors

Expanded text and reformatted data presentation have improved the quality of the manuscript. 

Author Response

Expanded text and reformatted data presentation have improved the quality of the manuscript. 

Response: We thank the Reviewer for appreciating our efforts.